# Collaborative Distributed Planning with Asymmetric Information. A Technological Driver for Sustainable Development

**Gregorio Rius-Sorolla ***[ID]**, Julien Maheut ***[ID]**, Sofia Estelles-Miguel and Jose P. Garcia-Sabater**

Departamento de Organización de Empresas, Universitat Politècnica de València, Camino de Vera s/n, 46022 Valencia, Spain; soesmi@omp.upv.es (S.E.-M.); jpgarcia@omp.upv.es (J.P.G.-S.)
* Correspondence: greriuso@upv.es (G.R.-S.); juma2@upv.es (J.M.)

**Abstract:** The growing interest in sustainable development is reflected in both the market's sensitivity to environmental and social issues and companies' interest in the opportunities that sustainable development objectives provide. SMEs, which account for most of the world's pollution, have significant resource constraints for a sustainable development. Sharing their scarce resources can help them to overcome these constraints and to gain agility and organisational resilience against uncertainties, but the distrust inherent in belonging to different companies prevents them from sharing the necessary information for coordination purposes. This paper presents a coordination mechanism proposal with information asymmetry to allow independent companies' resources to be sustainably shared as a technological driver. The proposed distributed coordination mechanism is compared to both a decentralised–uncoordinated and a centralised situation. The interest of the proposal is evaluated by a computer simulation experiment employing mathematical programming models with independent objectives in the Generic Materials and Operations Planning formulation with a rolling horizon procedure in different demand, uncertainty and product scenarios. Competitive improvement is identified for all members for their excess capacity use and their operations planning.

**Keywords:** supply chain planning; sustainability; lagrangian relaxation; resources sharing; collaborative planning; mathematical programming

## 1. Introduction

In the last decade, corporate interest in green investments has considerably increased, because companies are concerned about resource efficiency and environmental issues [1] and the private sector's commitment to collaborate [2]. This trend is a result of public policies. For example, one of the three main European Commission objectives for environmental policy is the decoupling of resource use from economic growth through significantly improved resource efficiency, dematerialisation of the economy and waste prevention [3]. Fulfilling this goal requires synergistic changes in both policy and industry terms [4]. The sustainability concept in the supply chain management field was introduced by Carter et al. [5]. Seuring et al. [6] define sustainable supply chain management (SSCM) as the management of material, information and capital flows and as cooperation among companies along the supply chain, while taking goals from all three sustainable development dimensions (economic, environmental, social) into account, which derive from customer and stakeholder requirements.

In order to achieve SSCM, the sustainable consumption and production topic is one of the most crucial aspects to consider. It consists of having more efficient and profitable production, using fewer raw materials and adding value to a product, while creating less pollution and waste during this process [7]. Tseng et al. [8] explain that SSCM reduces resources, material and waste by enabling better resource utilisation, which plays a significant role in achieving social, environmental and economic performance.

Industrial symbiosis is another strategy to achieve SSCM [9], which is the collective resource optimisation concept based on sharing services, utility and by-product resources

among diverse industrial processes or actors to add value, reduce costs and improve the environment. The keys to industrial symbiosis are the collaboration and synergistic possibilities offered by geographic proximity, which generally focuses on the physical exchange of materials, energy, water and by-products. Industrial symbiosis could be a considerable financial benefit in raw material substitution and transportation cost savings [10].

In the responsible production quest towards sustainability, SMEs (small- and medium-sized enterprises) are identified as a group that contributes to a large quantity of global pollution [11]. SMEs account for more than 99% of European enterprises, employ almost 70% of the European workforce and produce around 60% of overall manufacturing and services turnover [12]. SMEs present some characteristic barriers and drivers to engage in environmental management and resource efficiency. Their limited resources and knowledge, interest and motivation for environmental issues [13] are some of the main barriers. Nevertheless, research into sustainability in SMEs is limited [14]. Furthermore, original equipment manufacturers (OEMs) are also expected to manage and coordinate their activities in the supply chain to share and reduce resource use [15].

Sharing resources between companies is an action that overcomes limitations of available resources, an action that can improve economic performance and service levels and reduce the overall environmental impact. Moreover, efficiently used resources can enable the creation of productive employment. Sharing resources requires coordination and represents an opportunity for sustainable development in the supply chain [16]. In logistics, the potential of logistic-sharing solutions and respective transport capabilities to reduce emissions and mitigate the transport sector's impacts on climate change also implies benefits for companies by reducing overall operating expenses and transport costs per kilogram and by cutting maintenance and personnel costs, because fewer assets are needed [17]. As Shuai et al. [18] point out, online retailers usually adopt capacity sharing to cope with the demand surge because of unmanned distribution's low cost, especially because demand tends to be uncertain.

Sharing resources is increasingly easier, thanks to digitisation [19], regardless of the cooperation level, while organisations' increased resilience helps deal with the complexity of change, while preserving the capacity for development [20]. However, companies are reluctant to share their internal information, a requirement for a supply chain's centralised coordination [21]. Theoretically, centralised coordination through mathematical programming models ensures the possibility of reaching the cheaper solution on the whole and, consequently, the most sustainable operation planning by making the use of global resources efficient and sustainable independently of shared resources existing [22].

The twofold problem of sharing resources throughout coordinated operations planning across many supply chains and searching for a near-optimal solution with mathematical programming has been addressed in the literature since Ertogral et al. [23]. Coordination can be achieved with two decision-making approaches: centralised and decentralised [24]. Centralised coordination can accomplish optimal sustainable actions but requires each member to share their internal information with a central agent. Decentralised coordination entails designing and adopting coordination mechanisms [25] and should overcome lack of confidence in sharing internal information to become a technological driver of sustainable development. These coordination mechanisms are necessary to overcome an uncoordinated situation and to move closer to centralised coordination. Green technologies are considered an effective way of easing environmental pressure [26]. This paper focuses on the proposal of coordination mechanisms for capacitated operations planning within a supply chain where decisions are made through mathematical programming models, and members distrust sharing all information.

Operations alignment and improvement can be key drivers for sustainable development in companies with limited resources. This is a challenging task, because business' dynamic natures require constantly updating coordination decisions with time. Replanning can be triggered by specific events or periodically carried out. In the latter multiperiod case, the decisions made at the beginning of a period remain in place until the end of the

period and are reconsidered before the next period begins. Therefore, decisions are made under uncertainty, because perhaps not all information is known, or known with certainty, at the decision-making time. A recent literature review on coordination mechanisms for decentralised decision making [21] concluded that studies that cover the analysis of coordination mechanisms in multiperiod contexts and, more precisely, in decentralised settings with multiple independent decision makers were lacking [27].

This article contributes to shedding some light on designing and understanding a coordination mechanism that addresses lack of research attention to sustainability in SMEs. This article addresses the research question of a proposed coordination mechanism for distributed collaborative operation planning between independent companies to share resources with asymmetric information and to face demand uncertainty to outperform an uncoordinated situation and a centralised situation towards sustainable development. The coordination mechanism is used in decentralised multiperiod contexts with information sharing concerns as in independent SMEs. The coordination mechanism uses the convergence of Lagrangian multipliers that are updated by the subgradient method. This approach is applied to an extensive test bed that represents a cluster of companies with no prevailing power that voluntarily decide to share the capacity of one of their resources, e.g., transport or batch processing ovens (welding, annealing, vulcanisation, etc.), towards a more sustainable supply chain. To cope with the dynamic nature of demand, a multiperiod decision approach based on a rolling horizons procedure [28] is proposed. To model supply chain decisions, the Generic Materials and Operations Planning (GMOP) formulation is used, because it considers alternative operations and an alternative bill of materials in a compact manner [29]. To the best of our knowledge, this is the first research study to use the GMOP modelling approach to integrate distributed coordination mechanisms in the search for a more sustainable supply chain. It is a work continuation started by Rius Sorolla's thesis [24].

The remainder of the paper is structured as follows: firstly, an introduction to the coordination mechanism is presented; secondly, the research methodology is followed; thirdly, the proposed coordination mechanism is applied with GMOP formulation; fourthly, some numerical experiments are performed, and the results are discussed; finally, some conclusions and future works are provided.

## 2. Coordination Mechanism Review

Coordination mechanisms can be grouped into auctions, hierarchy, metaheuristics and mathematical decomposition [21]. Auctions are important coordination mechanisms that have been employed ever since the earliest times to allocate goods and services [30]. Dash et al. [31] use a Continuous Double Auction as an extension of the Vickrey–Clarke–Groves auction, where agents are encouraged to honestly report their capacity and costs. Other auctions utilise the nonlinear mathematical programming modelling approach [32]. Extensive literature in auction theory can be found in [33]. However, according to Mason et al. [34], auctions present susceptibility to collusion and other pathologies that render it undesirable in practice. Moreover, auction mechanisms imply low convergence speed in computational efficiency [35] to identify a sustainable solution.

Hierarchical coordination can be understood as an extension of auctions, where only one of the parties bids. The silent bidder tends to obtain better terms than the bidder [30] that acts as a barrier towards sustainable coordination. It generates better results than the uncoordinated situation does. The hierarchical coordination mechanism can be initiated through downstream proposals without subsequent negotiations [36]; upstream [37,38]; and with counterproposals [39], negotiations [40] and compensations [41]. Nevertheless, hierarchical mechanisms do not consider the effects of decisions on all partners, locally [42]. Dudek and Stadtler [22] compare upstream planning to centralised planning and observe 14.1% differences in total supply chain costs on average.

Another alternative to speed up searches among alternatives is to use metaheuristics or mathematical decomposition tools as coordination mechanisms. Metaheuristic

coordination mechanisms can use the ant colony [43], simulated annealing [22,44–47], neighbourhood search [48], genetic algorithms [49] or immune systems [50]. They all allow searches for better coordination according to preestablished rules, but do not guarantee optimality or knowing the assumed gap. Mathematical decomposition can be grouped in Dantzig–Wolfe's method [51], Benders' method [52] and other Lagrangian decompositions. Dantzig–Wolfe's method requires a centralised mediator to update the internal price assigned to the use of the shared resource [34,53] with a view to seeking to comply with relaxed constraints. Therefore, a centralised mediator must have access to all information. Benders [52] proposes dualising the objective function and then relaxing constraints to generate subproblems or separable agents, where a centralised problem must add the optimal value of each subproblem to the constraints and functions that cannot be decomposed [54]. This master–slave structure can be found in Dantzig–Wolfe's [51] and Benders' decompositions [52], among others. With Lagrangian decomposition, a structured decomposition of the problem is used to achieve coordination. Lagrange multipliers are generally updated with the subgradient method [55–58], as pointed out by Rius-Sorolla et al. [59].

Lagrangian relaxation can be considered a very efficient coordination tool, because it decomposes supply chain decisions into a set of related subproblems. Actually, decomposition enables the centralised decisional problem to be broken down into a series of independent subproblems with decisions coordinated through the master problem [60]. The coordination of the Lagrangian relaxation subproblem can be accomplished without a master problem, as proposed by Singh et al. [61]. In their method, referred to as the "safe multipart computing procedure", information is shared without disclosing which supply chain member shared data come from. Therefore, each member can calculate penalties for using the shared resource by the subgradient method. The subgradient method offers conspicuously rapid convergence for Lagrange multipliers [59] that is, however, erratic in the proposal convergence for the main function.

The Lagrange multiplier allows certain constraints to be eliminated or relaxed in return for penalising noncompliance in the objective function to, thus, simplify the problem. The new objective function that includes penalties for the eliminated constraints is called the relaxed function [62]. The optimal relaxed function value provides the main problem with a lower bound (minimisation objectives) for a given nonnegative Lagrange multiplier value, because it only adds a negative term to the objective function [55]. The relaxation of certain constraints may allow a dual model to be generated that can be more easily resolved [58].

The Lagrange multiplier method allows a problem to be decomposed into a series of coordinated subproblems. In fact, by relaxing certain constraints, some parts of the main function and constraints that become subproblems can be independently solved. These subproblems are linked together by the parts of the main function and constraints that cannot be decomposed. These subproblems can be considered independent entities [51] that are coordinated by a central entity with which they share certain information [34]. The nondecomposed parts of the main function allow new values for the Lagrange multipliers to be generated. Optimal planning for subproblems can be found from the new values of the Lagrange multipliers, and new limits to the main problem can be generated. However, the relaxation of these constraints can generate completely independent subproblems that are coordinated only by Lagrange multipliers. In these cases, no further decomposition can take place, and the method is relied on to find the most appropriate values for the Lagrange multipliers to coordinate subproblems. The method must enable the best solution proposals for each subproblem to simultaneously involve the best solution proposal for the global problem and, at the same time, comply with relaxed constraints.

The subgradient method requires not only information about the optimal value of each subproblem to be shared but also the penalty to be applied for breaching relaxed constraints and the optimal planning cost without applying the penalty [58]. In other words, entities must share the result of their local optimisation and compliance with the relaxed or shared constraints for each Lagrange multiplier value.

Regarding information exchange, a third independent agent can be put in place to collect all the information and recalculate Lagrange multipliers [60]. Alternatively, as only aggregate information is needed to calculate the Lagrange multiplier, a secure information exchange protocol can be established [61]. For example, the first entity informs the other entities about the required values by adding a random amount. Then, all the other entities add their local data to these to generate the aggregate of all the entities. Subsequently, the first agent subtracts the random amount when the final data are returned with all the entities' local data included [61]. In addition, if information is available for all the agents at the same time (i.e., an iterative distributed decision-making process), no independent coordinating agent is required [50]. Notwithstanding, a new local constraint can be added for certain problems, which facilitates the selection of local planning to comply with relaxed constraints [63].

## 3. Methodology

A multistep approach was followed at different demand uncertainty levels to help to improve operations planning for sustainable development.

Firstly, the GMOP formulation is presented. It helps to establish mathematical programming that contemplates the possibility of alternative operations [29,64,65]. Therefore, the adopted formulation moves closer to companies' reality.

This section presents the modelling of the centralised coordination model through the compact model and how the model formulation also allows uncoordinated models to be considered. Afterwards, the proposed coordination mechanism that uses Lagrange relaxation is introduced. The Lagrangian multipliers calculation, which allows prices to be obtained for shared resources and the distributed coordination mechanism, is described. A flow chart of Figure 1 presents the modelling approach, Figure 2 the experiment design, while Figure 3 will present the coordination mechanism proposal steps.

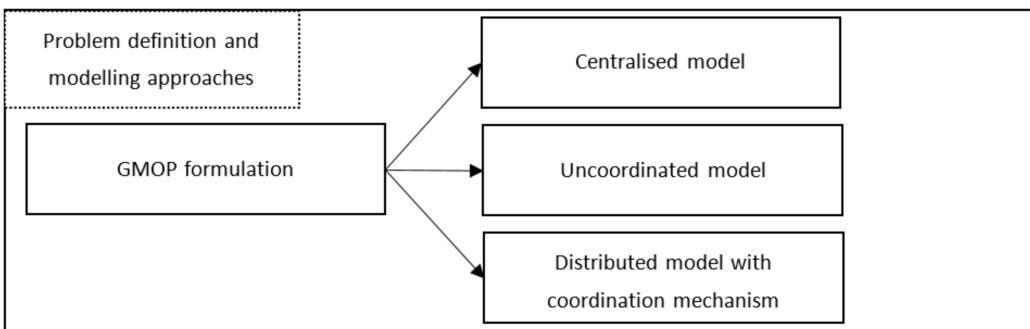

**Figure 1.** Modelling approaches.

Secondly, the design of the numerical experiment is presented based on an available test bed [66]. This test bed allows several operations planning variables to be analysed, such as different demand types including expected variations (constant, trend, seasonal, combinations), suspected variations (noise in uniform distribution), unknown variations (uncertainty with random variation) and the irregular distribution of demand among final products. This test bed provides different product complexities and the possibility of alternative operations. These instances include variations across periods and between several planning horizons, which takes us closer to the reality of companies' uncertainties. The test bed takes into account the need to break symmetries to facilitate resolution and to avoid instabilities in its calculation [67]. The cost structure varies according to the proximity to the decision making in each planning horizon by adjusting to the industrial reality of considering resource variations in mid- and long-term forecasts. Resource capacities have different availability levels that facilitate the analysis of instability situations, because the companies' available capacities are saturated. Instances are designed to use the rolling

horizon heuristic present in industry [68] with previous periods to avoid initial stocks and final periods influencing the simulating continuity objective.

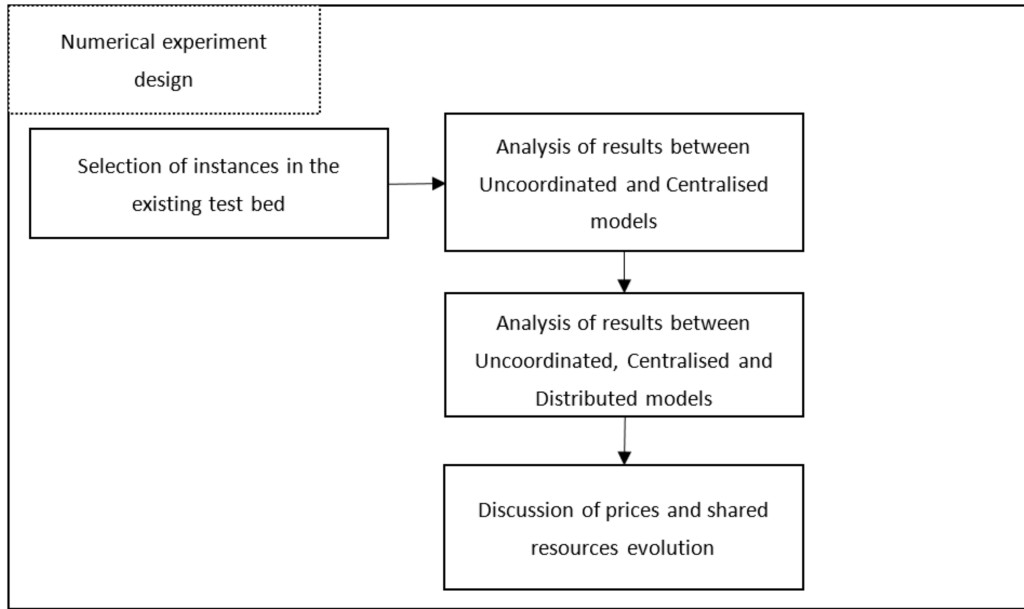

**Figure 2.** Numerical experiment design.

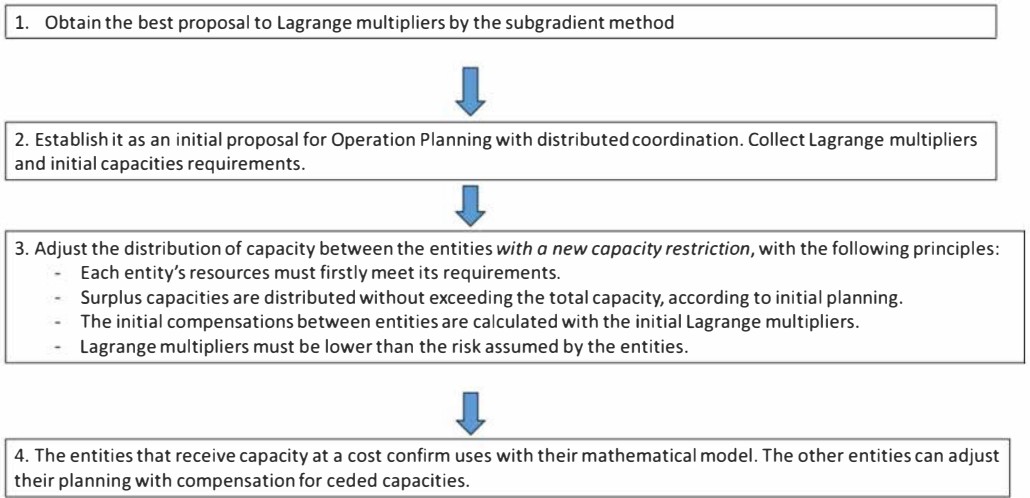

**Figure 3.** Flowchart of the proposed coordination mechanism (Source: the Authors).

For this computer simulation, 216 instances were selected to analyse the influence of various saturation levels of available resources and the impact of three product structures. These instances are considered the most susceptible, because they involve using more of their resource capacities and, consequently, have more probabilities of requiring shared resources. The variables of unknown demand variation and irregular demand distribution have been left for future analysis work.

In order to evaluate different solutions, both total costs and service levels were taken as indicators of economic, environmental and social sustainability. It was assumed that employing the available resources between companies would lead to better environmental performance and cost reduction towards economic sustainability and that both would create jobs.

Thirdly, the simulation results of mathematical programming in an uncoordinated situation versus the operation planning of centralised coordination are presented. This comparison allows those cluster families for which available capacity is lacking to be

identified, and it is possible to share their resource capacity, which could be a competitive advantage. Subsequently, in those families of instances for which improvements were observed, distributed operations planning was simulated with the proposed coordination mechanism. This allowed the results of the total costs, service levels and number of periods sharing resources for each instance family to be analysed.

## 4. Proposed Coordination Mechanism with GMOP Formulation to Model Multisite and Multiproduct Operations Planning for the Sustainable Resource Sharing of Independent Companies

The proposed coordination mechanism follows Figure 3. Its steps are done following the GMOP formulation.

GMOP is a multisite, multiproduct, multiprocess, multiperiod and multiresource operations planning formulation that includes the operations or *stroke* decisions. Therefore, this formulation allows the representation of parallel processes, coproducts [69,70], alternative processes [71,72] and other possibilities that are inherent to the use of *strokes* that make it easier to represent problems than Gozinto-based structures [73]. The *stroke* decision enables limited environmental resources to be considered and the lowest impact processes and the highest sustainability to be chosen. The GMOP formulation herein presented includes an index to discriminate the supply chain companies to enhance the clearness of equations. Table 1 lists the indices, parameters and variables.

**Table 1.** The indices, parameters and variables used in the GMOP formulation.

| Indices | |
|---|---|
| $i$ | Index set of SKUs (including products, packaging and site) |
| $t$ | Index set of planning periods in each PH (t′ refers to the total studied horizons) |
| $r$ | Index set of resources |
| $k$ | Index set of *stroke*s |
| $ro$ | Index set of each Planning Horizon (PH) |
| $c$ | Index set of each company |
| $j$ | Index set of Lagrange iteration |

| Parameters | |
|---|---|
| $D_{i,t,\text{ro},c}$ | Demand for SKU *i* for period *t* to company *c* on PH *ro* |
| $H_{i,t,c}$ | Cost of storing one unit of SKU *I* during period *t* at company *c* |
| $CO_{k,t,c}$ | Cost of *stroke k* during period *t* at company *c* |
| $CS_{k,t,c}$ | Cost of setting up *stroke k* during period *t* at company *c* |
| $CB_{i,t,c}$ | Cost of delay of SKU *I* during period *t* at company *c* |
| $SO_{i,k,c}$ | Number of units of SKU *i* that generates a *stroke k* at company *c* |
| $SI_{i,k,c}$ | Number of units of SKU *i* that *stroke k* uses at company *c* |
| $LT_{k,c}$ | Lead time of *stroke k* at company *c* |
| $KAP_{r,c}$ | Capacity availability of resource *r* during period *t* at company *c* (in time units) |
| $KAP_{r=1,c}$ | Capacity availability of shared resource *r* during period *t* at company *c* (in time units) |
| $KAP'_{r=1,c}$ | Capacity availability with coordination of shared resource *r* during period *t* at company *c* (in time units) |
| $M_c$ | A sufficiently large number at company *c* |
| $TO_{k,r,c}$ | Capacity of resource *r* required to execute one unit of *stroke k* at company *c* (in time units) |
| $TS_{k,r,c}$ | Capacity required of resource *r* for setting up *stroke k* at company *c* (in time units) |

| Variables | |
|---|---|
| $z_{k,t,\text{ro},c}$ | Amount of *strokes k* to be *k* to be performed during period *t* on PH *ro* at company *c* |
| $\delta_{k,t,\text{ro},c}$ | =1 if *stroke k* is performed during period *t* on PH *ro* (0 otherwise) at company *c* |
| $f_{i,t,\text{ro},c}$ | Delay quantity of SKU *i* during period *t* on PH *ro* at company c |
| $x_{k,t,\text{ro},c}$ | Stock level of SKU *i* on hand at the end of period *t* on PH *ro* at company c |

*SKU* Stock-Keeping Unit, *PH* Planning Horizon. (Source: based on Garcia-Sabater et al. [29]).

The GMOP formulation is then stated as:

$$min \sum_t \sum_i (H_{i,t,c} x_{i,t,ro,c} + CB_{i,t,c} f_{i,t,ro,c}) + \sum_t \sum_k (CS_{k,t,c} \delta_{k,t,ro,c} + CO_{k,t,c} z_{k,t,ro,c}) \forall ro, c \quad (1)$$

So that:

$$x_{i,t,ro,c} = x_{i,t-1,ro,c} - D_{i,t,ro,c} + f_{i,t,ro,c} - f_{i,t-1,ro,c} - \sum_k (SI_{i,k,c} z_{k,t,ro,c}) + \\ \sum_k (SO_{i,k,c} z_{k,t-LT_k,ro,c}) \forall i, t, ro, c \quad (2)$$

$$\sum_k (TS_{k,r,c} \delta_{k,t,ro,c}) + \sum_k (TO_{k,r,c} z_{k,t,ro,c}) \leq KAP_{rc} \forall r, t, ro, c \quad (3)$$

$$z_{k,t,ro,c} - M_c \delta_{k,t,ro,c} \leq 0 \ \forall k, t, ro, c \quad (4)$$

$$x_{i,t,ro,c} \geq 0; \ w_{i,t,ro,c} \geq 0, \ \forall i, t, \text{ro}, \ c; \ z_{k,t,c} \in \mathbb{Z}^+; \ \delta_{k,t,c} \in \{0,1\} \ \forall k, t, ro, c \quad (5)$$

In the centralised GMOP formulation, the purpose of the objective Function (1) is to minimise the sum, for all the companies, of inventory costs, penalties for service delays, setup costs and operations costs. Equation (2) ensures a stock balance at each company and logistics connections (stock, delays, demand) with operations (component use and new product operations). Equation (3) defines the limitation of the available resources at each company for each period. Equation (4) establishes setup requirements when products are manufactured during period t with *stroke* k by each company. Finally, Equation (5) establishes the range for variables.

It is considered that one of the resources (i.e., resource 1) can be shared by all the companies. To this end, and to compute the sum of the total costs incurred by each company, the objective function is rewritten as Equations (6) and (3), which limit resources' capacities, and are replaced with Equations (7) and (8), respectively.

$$\sum_c F_c = min \sum_c \left( \sum_t \sum_i (H_{i,t,c} \cdot x_{i,t,ro,c} + CB_{i,t,c} \cdot f_{i,t,ro,c}) \\ + \sum_t \sum_k (CS_{k,t,c} \cdot \delta_{k,t,ro,c} + CO_{k,t,c} \cdot z_{k,t,ro,c}) \right) \forall ro \quad (6)$$

$$\sum_c \left( \sum_k (TS_{k,r=1,c} \cdot \delta_{k,t,ro,c}) + \sum_k (TO_{k,r=1,c} \cdot z_{k,t,ro,c}) \right) \leq \sum_c KAP_{r=1,c} \ \forall t, ro \quad (7)$$

$$\sum_k (TS_{k,r,c} \cdot \delta_{k,t,ro,c}) + \sum_k (TO_{k,r,c} \cdot z_{k,t,ro,c}) \leq KAP_{r,c} \ \forall t, c, r \neq 1, \text{ro} \quad (8)$$

By doing so, we find that Equation (7) is the only one that cannot be separated by the company index, because it simultaneously concerns all the companies. Therefore, the Lagrange decomposition approach is applied to this equation, which is relaxed and moved to the objective function by means of Lagrange multipliers (or penalties). This leads to Equation (9).

$$\max_{u_{t,ro}^j} min \sum_c \left( \sum_t \sum_i (H_{i,t,c} \cdot x_{i,t,ro,c} + CB_{i,t,c} \cdot f_{i,t,ro,c}) \\ + \sum_t \sum_k (CS_{k,t,c} \cdot \delta_{k,t,ro,c} + CO_{k,t,c} \cdot z_{k,t,ro,c})) + \\ + \sum_t u_{t,ro}^j \left( \sum_c \left( \sum_k (TS_{k,r=1,c} \cdot \delta_{k,t,ro,c}) \\ + \sum_k (TO_{k,r=1,c} \cdot z_{k,t,ro,c})) - \sum_c KAP_{r=1,c} \right) \forall ro \quad (9)$$

The centralised formulation can now be decomposed into *c* models, where one is related to each company, according to Equation (10) and Constraint (11), along with Equations (2), (4) and (5). Lagrange multipliers act as a common coordination mechanism

to all the models. Indeed, a given set of values for Lagrange multipliers allows optimal planning for each company.

$$
\begin{aligned}
F_{cd} = \max_{u^j_{t,ro}} \min \sum_t \sum_i (H_{i,t,c} \cdot x_{i,t,ro,c} + CB_{i,t,c} \cdot f_{i,t,ro,c}) \\
+ \sum_t \sum_k (CS_{k,t,c} \cdot \delta_{k,t,ro,c} + CO_{k,t,c} \cdot z_{k,t,ro,c}) \\
+ \sum_t u^j_{t,ro} \left( \sum_k (TS_{k,r=1,c} \cdot \delta_{k,t,ro,c}) + \sum_k (TO_{k,r=1,c} \cdot z_{k,t,ro,c}) \right. \\
\left. - KAP_{r=1,c} \right)
\end{aligned}
\tag{10}
$$

$$
\sum_k (TS_{k,r,c} \cdot \delta_{k,t,ro,c}) + \sum_k (TO_{k,r,c} \cdot z_{k,t,ro,c}) \leq KAP_{r,c} \ \forall t, c, r \neq 1, ro
\tag{11}
$$

Nevertheless, finding the set of multipliers that leads to the global optimal solution can be a challenging task. The subgradient method presents the fast convergence of the relaxed function and Lagrange multipliers but not in the main function [59]. Therefore, $T$ multipliers are calculated from Equation (12), where the positive value is applied by adjusting the previous multiplier, plus the breach of the relaxed constraint affected by step $s^j$, which is calculated by Equation (13).

$$
\begin{aligned}
u^{j+1}_{t,ro} = \max \Big\{ 0, u^j_{t,ro} \\
+ s^j_{t,ro} \left( \sum_c \left( \sum_k (TS_{k,r=1,c} \cdot \delta_{k,t,ro,c}) + \sum_k (TO_{k,r=1,c} \cdot z_{k,t,ro,c}) \right) \right. \\
\left. - \sum_c KAP_{r=1,c} \right) \Big\} \forall \ t, ro
\end{aligned}
\tag{12}
$$

$$
s^j_{t,ro} = \frac{\sigma^j_{ro} (F^* - F_D(u^j))}{\sum_t \| (\sum_c (\sum_k (TS_{k,r=1,c} \cdot \delta_{k,t,ro,c}) + \sum_k (TO_{k,r=1,c} \cdot z_{k,t,ro,c})) - \sum_c KAP_{r=1,c}) \|^2} \forall \ ro
\tag{13}
$$

where $\sigma^j_{ro}$ is a scalar that must satisfy $0 < \sigma^j_{ro} \leq 2 \ \forall j, ro$. $F^*$ is the aggregate of the lowest planning cost of all the entities that complies with the relaxed constraints, i.e., the main function. $F_D(u^j)$ is the value of the planning cost of all the entities penalised by the gap between the shared capacity constraints and Lagrange multipliers $u^j_{t,ro}$, i.e., the relaxed function. Fisher [55] recommends starting with $\sigma^0_{ro} = 2$ and reducing by half if $F_D(u^j)$ does not improve the lower bound after 10 iterations.

In order to improve convergence, the addition of new constraints (Equation (14)) to the shared resources in each company is proposed. These constraints should be based on the best available relaxed function solution obtained from the stabilisation of Lagrange multipliers in their updated cycle.

$$
\sum_k (TS_{k,r=1,c} \cdot \delta_{k,t,ro,\ c}) + \sum_k (TO_{k,r=1,c} \cdot z_{k,t,ro,c}) \leq KAP'_{r=1,c} \ \forall t, ro, \ c
\tag{14}
$$

In the companies that do not use the complete capacity of their available resource, the new capacity limitation equals the capacity employed by the company in accordance with the proposed solution selected as the basis for Lagrange multipliers' stabilisation. The companies with needs that exceed their capacity receive greater capacity from other entities' spare capacities. Surplus resources are distributed proportionally to the several proposed resource uses in the best available solution of the aggregate relaxed function; see Equation (10).

These resource capacity constraints firstly ensure local needs according to the initial solution and then offer their excess capacity of the shared resource to other entities in proportion to the volumes they request in the initial solution. This distribution is understood to occur only when at least one entity has excess available capacity of the shared resource during this period, and at least one other entity requires an amount of it that

exceeds its own capacity during the same period. The total sum of the new requirements is less than or equal the total capacity of the available resource. The entities that need additional capacity of the shared resource should recalculate their master plan according to the additional resource capacity available for these entities, KAP′, during each period in which the resource is shared.

Lagrange multipliers can be understood as the unit cost for borrowing shared resource capacity. By multiplying Lagrange multipliers by the capacities used by each entity, they can be understood as the compensation that entities must provide for the right to use additional capacities. This penalty cost refers to the right to use and not to the use itself and is understood to be made by each company that employs the shared resources capacities in this formulation.

The values that are selected for Lagrange multipliers must also be considered. The amounts assigned to each company are modified from the total available resources from a breakeven point. The result may vary in relation to the initial total costs. However, feasibility is guaranteed by ensuring that the use of the available resource capacity is not exceeded. This procedure makes the most of the way that the subgradient method stabilises Lagrange multipliers compared to other updated methods. Noncompliance with the relaxed constraints of the subgradient method is also overcome by forcing a proportional distribution of resources according to the best proposal in Lagrange multipliers stabilised by the subgradient method.

In this new situation, the entities that need more resources during specific periods can access greater availability of resources, depending on those used by other entities, but can be penalised by the permitted penalties. However, entities can perform their operations planning accordingly and can determine the final use that they apply to the additional resource in question.

The entities that reserve some resources to be allocated to other entities can replan when the other entities confirm the capacity they intend to use from the shared resource and the compensation they will pay for the right to use this shared resource.

## 5. Numerical Experiments

In order to validate the proposed resource sharing mechanism, a large instance test bed [66] is used to compare each entity's total costs and service level among (1) centralised coordination, (2) an incoordination situation and (3) distributed coordination proposals by a rolling horizon procedure. Total costs are defined as the sum of the costs of the 52 analysed executed periods. The service level is defined as the unmet demand level according to the demand requirements during the 52 studied periods, according to Equation (15) [74], where $f_{i,t,ro}$ is the amount of product $i$ in the delay during period $t$ on planning horizon $ro$, and $D_{k,t,ro}$ is the demand of product $i$ during period $t$ on planning horizon $ro$. The executed period is only during $t$ and equals 1 as the replanning period and frozen interval are chosen as 1. These experiments are used to validate the expected hypothesis that distributed coordination approaches centralised coordination and improves the incoordination situation.

$$NSR = \frac{\sum_{ro=13}^{64}\left(1 - \frac{\sum_{t=1}^{FI}\sum_i f_{i,t,ro}}{\sum_{t=1}^{FI}\sum_i D_{i,t,ro}}\right)}{52} \tag{15}$$

The test bank has 4320 possible combinations, with 12 instances in each case (Table 2). Each instance has 71 demand periods, with demand updates during each period and 8 planning horizon periods. Demand differs for all the 10 final products in each instance. The 71 demand periods consist of 12 initial periods, when initial stocks are guaranteed to not influence the following 52 planning and analysis periods. Moreover, instances have 7 final demand periods to simulate continuity in planning. The instances are available online: http://personales.upv.es/greriuso/TEST_BED_GMOP.rar (accessed on 15 December 2020).

**Table 2.** Instance parameters.

| Pareto | Demand | Uncertainty | BOM/BOP | Saturation | Instance |
|---|---|---|---|---|---|
| Par00, Par05, Par10, Par15, Par20, Par25 | CC, TT, SS, ST, SD | CV00, CV10, CV20, CV3, CV40, CV50 | P1, P2, P3, P4, P5, P6 | R00, R75, R50, R30 | 1, 2, 3, 4, 5, 6, 7, 8, 9, 10, 11, 12 |

Source: the Authors.

P1, P4 and P6 are the selected product structures, as shown in Figure 4. The 10 final products have the same structure in each instance. Each product is the result of its *strokes* family, including those that use components and resources, and others related to purchases (highlighted in grey) that do not use entity resources. The end products of structure P6 have alternative processes. Only a relatively few instances are selected for this paper. The remaining instances are reserved for future research to cover other independent factor combinations.

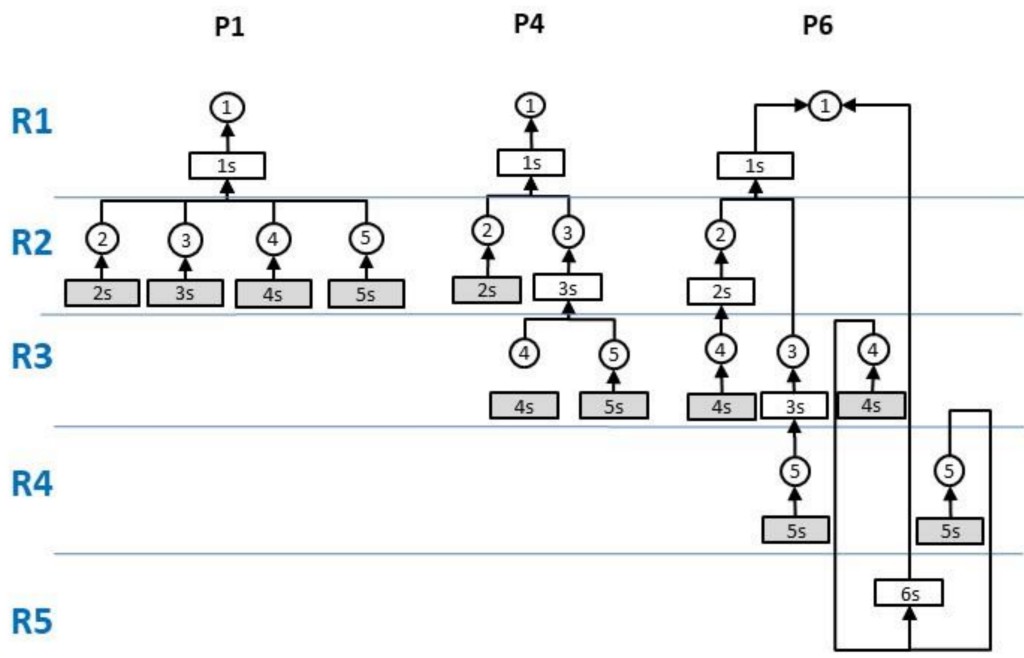

**Figure 4.** Bill of materials and process of different product types (Source: the Authors).

Demand is distributed among the 10 final products according to Pareto factor *Par00*, where all the products in each instance have similar requirements, except for the random parameters introduced in both demand noise and uncertainty. Both increasing demand *TT* cases and the pattern of increasing seasonal demand *ST* are selected (see Table 3). Demand is made up of the average demand for $\mu_t$ for each period $t$, set at 500 units; noise $Z_t$, calculated by a random uniform function of type $+/-$ 5 units; and a linear increasing function with constant slope $B_t$.

**Table 3.** Types of used demand.

| Demand Type | Function |
|---|---|
| Trend (TT) | $D_t = \mu_t + B_t + Z_t$ |
| Seasonal + Trend (ST) | $D_t = \mu_t (1 + \sin(2\pi t/52 + \pi/2)) + B_t + Z_t$ |

$\mu_t$ average demand, $B_t$ demand with a constant increase slope, $Z_t$ noise.

Demand uncertainty is one of the most important factors in supply chain instability [75]. Uncertainty is determined as having a 10% standard deviation over average period

demand *CV10*. Uncertainty is simulated as a normal random variation [76] that focuses on the demand for each product during each rolling horizon replanning period.

Finally, resource limitation influences the effects produced by planning horizon variations and frozen periods [77]. Figure 3 shows the different resources available for each *stroke* type. In this paper, resource availabilities of 30% (R30), 70% (R70) and 100% (R00) are selected for all the resources. In all, 216 instances are used to verify the proposed method. All the selected parameters are found in Table 4.

**Table 4.** Selected instance parameters.

| Pareto | Demand | Uncertainty | BOM | Saturation | Instance |
|---|---|---|---|---|---|
| Par00 | TT, ST | CV10 | P1, P4, P6 | R30, R75, R00 | 1, 2, 3, 4, 5, 6, 7, 8, 9, 10, 11, 12 |

Source: the Authors.

Instances are solved with a tolerance specification of a 1% allowed gap in a single compute core and with a processing limit of 3000 s for each rolling horizon. The exact resolution of the instances with the 71 periods is ruled out because after 120 h, the gap is still over 29.8%. Therefore, the rolling horizon heuristic procedure is applied to model the problem, which is common in both industry and academic terms [69]. Execution is carried out in the Rigel cluster based on the grid architecture and a multicore PC at the Universitat Politècnica de València with 72 Fujitsu BX920S3 nodes installed in the BX900S2 chassis. Each node includes two Intel Xeon E5-2450 processors (8 cores/16 threads, 2.1–2.5 GHz) and 64 GB of DDR3 RAM. Nodes are linked by two 10 GB Ethernet interfaces. The cluster runs a CentOS 6 operating system, and a Sun Grid Engine manages load. The multicore PC runs a CentOS 6.4 operating system by an Intel Core i5-4670 processor (4 cores/4 threads, 3.4 GHz) with 8 GB of DDR3 RAM [78].

The rolling horizons are defined with an eight-period planning horizon, one frozen period and one replanning period. Only the first period is executed. The plan for each period is updated based on both the results of the previous planning execution and the update of the demand forecast for the eight new periods. The results of the executed period become the starting point on the following rolling horizon: initial stocks, delays and orders launched. Programming is done in C#. The parameters of each entity are assigned according to the data collected in instances. The search for the best operational planning proposal on each rolling horizon is done with the GUROBI® 7.0.2 64 bits optimiser for Linux, because of its superior performance [79].

The first resource of the three independent entities is established as shared (R = 1). This selection can be made with either other shared resources or more resources, but for easy mathematical representations, the first one is selected. It is also assumed that each instance of the test bed collects the data that defines each entity. To run these numerical experiments, instances are alphanumerically selected so that TTP1R00_1, TTP1R00_2 TTP1R00_3 are the three instances from which the coordinated planning proposal is sought.

### 5.1. Centralised Coordination Resolution and Incoordination Resolution

The proposals are evaluated in relation to the centralised coordination of the three entities. Moreover, the separate resolution of each instance is assumed to be noncoordinated or decentralised–uncoordinated and to have a benchmark to be compared to the decentralised coordination mechanism. Firstly, each instance is solved separately, which is called the uncoordinated situation. Secondly, instances are solved in an attempt to minimise the aggregate costs of the three instances during the eight periods of each planning horizon by executing only the first period with centralised coordination, where the three entities share the capacity of their first resource. A comparison of the results between decentralised-uncoordinated and centralised coordination is found in Figure 5 for the total costs during the 52 analysed periods. Figures 5 and 6 present the total costs and Figures 7 and 8 the service levels. The service level in the 52 analysed periods is found in

Figure 7 for cases with trend demand "TT". Figures 6 and 8 depict the cases with seasonal trend demand "ST".

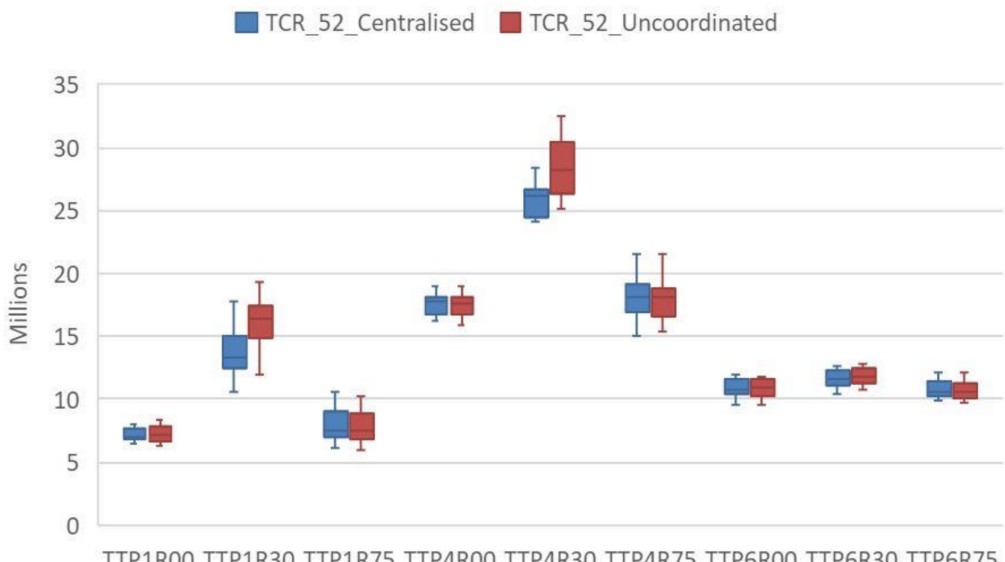

**Figure 5.** The total costs for decentralised–uncoordinated and centralised coordination for the 52 studied periods, in which all three entities share the capacities of the first resource with trend demand (the TTP1R00 trend demand, product type P1, resource at 100).

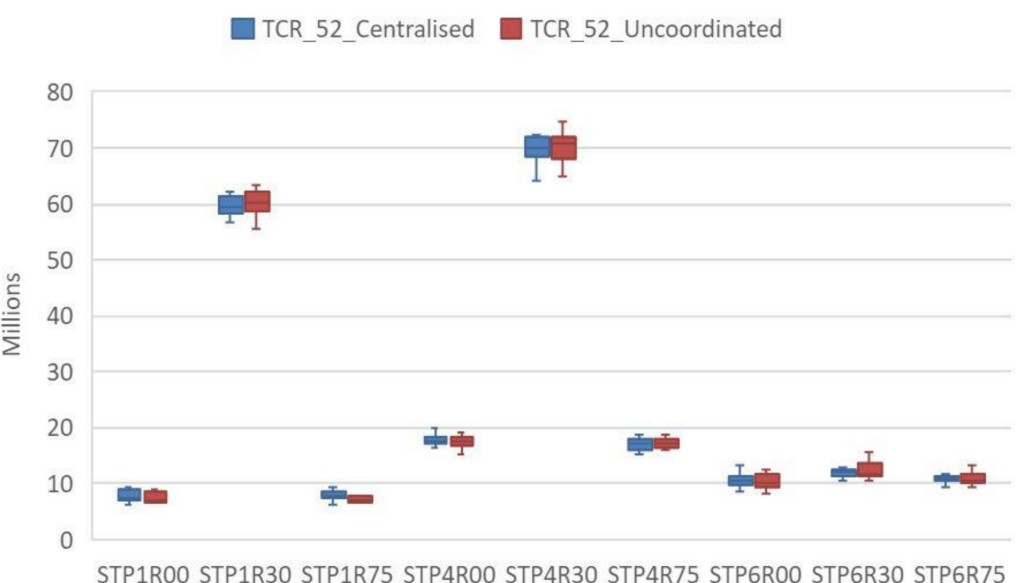

**Figure 6.** The total costs for decentralised–uncoordinated and centralised coordination for the 52 studied periods, in which all three entities share the capacities of the first resource with seasonal trend demand (the STP1R00 seasonal trend demand, product type P1, resource at 100%; the TCR_52_Centralised total cost values for the 52 studied periods with the entities sharing a resource in centralised coordination; the TCR_52_Uncoordinated total cost for the 52 studied periods with decentralised uncoordinated entities).

Figures 5 and 6 indicate the distribution of the total costs of the selected instances. Figures 7 and 8 denote the service level of the 12 instances of each resolved combination. Capacity constraints *R75* have no appreciable effect in relation to having 100% of resources *R00*. Required demands can be met with the available resources and there is no advantage in sharing resource *R1*. However, it should be noted that if the available resource is *R30* and product structures are *P1* and *P4*, sharing *R1* resources among the entities leads to a

saving in the total cost distribution median in Figure 5 and an improved service level in Figure 7. The between-pairs evaluation of the *TTP1R30* values, made by employing a signs test of their centralised and uncoordinated medians, shows that the null hypothesis can be rejected with reliability above 95% (p-Value = 0.00937). Nonetheless, the entities with *P6* product types have alternative processes that can compensate the reduction in resources at *R30.* Thus in these cases, no benefits come from sharing *R1* resources.

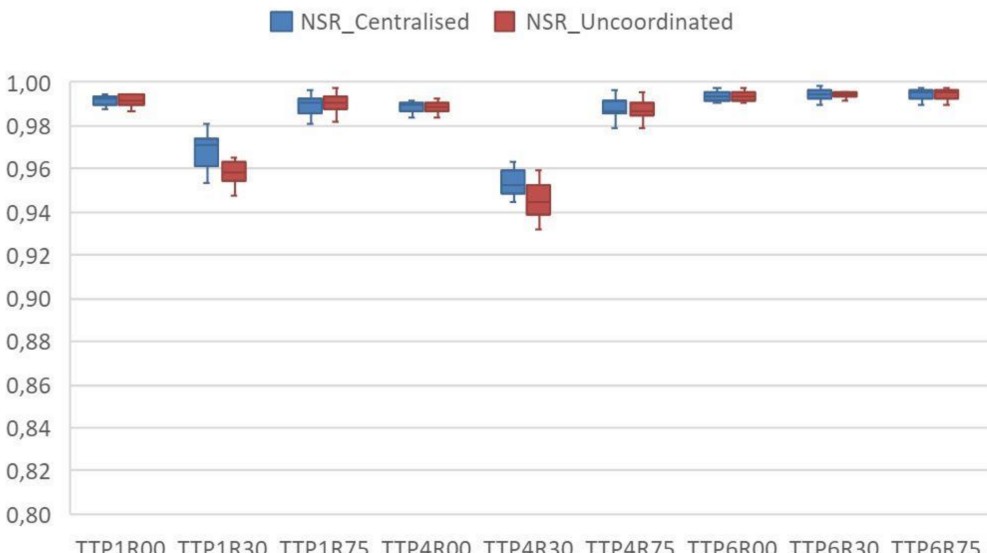

**Figure 7.** The service level of decentralised–uncoordinated and centralised coordination during the 52 studied periods, in which all three entities share the capacities of the first resource with trend demand (the TTP1R00 trend demand, product type P1, resource at 100%; NSR_Centralised Service level for the 52 studied periods with the entities sharing a resource in centralised coordination; the NSR_Uncoordinated service level for the 52 studied periods with decentralised uncoordinated entities).

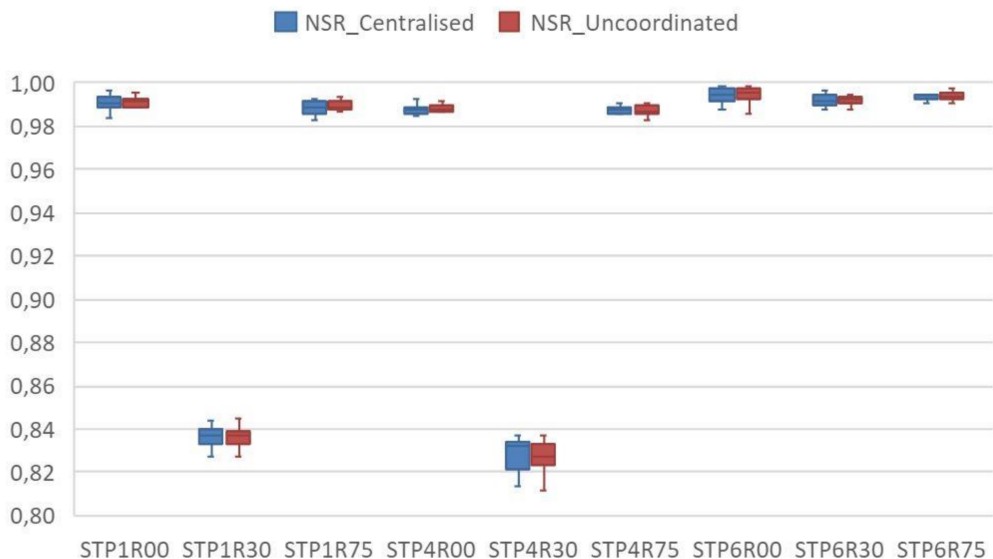

**Figure 8.** The service level for decentralised–uncoordinated and centralised coordination for the 52 studied periods, in which all three entities share the capacities of the first resource with seasonal trend demand (the STP1R00 seasonal trend demand, product type P1, resource at 100%; the NSR_Centralised Service level for the 52 studied periods with the entities sharing a resource in centralised coordination; the NSR_Uncoordinated service level for the 52 studied periods with decentralised uncoordinated entities).

Therefore, no average cost reduction takes place from the centrally shared capacities, except in those cases with combinations TTP1R30, STP1R30, TTP4R30 and STP4R30. In these cases, both the total costs and service level improve compared to the uncoordinated process. Only these combinations are selected for the comparison and analysis of the centralised, uncoordinated method and the proposed method for distributed coordination.

### 5.2. Distributed Coordination Resolution for Sustainable Resource Sharing of Independent Companies

As in centralised coordination, three entities share the capacities of the first resource, R1, in this case. These entities are coordinated by an internal unit penalty for the right of using the shared resource, established as a distributed coordination mechanism.

The operation seeks to move closer to possible real-world industrial circumstances. Firstly, entities inform one another about the resources they expect to require or that are surplus to their needs during each period, together with the differential or interest between their uncoordinated situation and the situation in which they have excess resources. Secondly, when this information becomes available, all the entities can calculate the Lagrange multiplier or the penalty for the right to use resources during each period. Subsequently, entities can update their situations in regards to the additional resources they need or that are available in excess during each period with the established penalties. Entities can also update the differential between their uncoordinated situation and the situation of having excess capacity, but with the penalties included. These steps are repeated until either a stable penalty situation is reached for the shared resource during all the planning periods on each rolling horizon or 300 Lagrange multiplier updating iterations are done. Figure 9 shows the Lagrange multipliers update on the 49th rolling horizon for the shared resource and for the eight planning horizon periods. The penalties of periods seven and eight are zero on the 49th planning horizon. Thirdly, the penalties and the initial distribution of resources that generate the lowest differential of the aggregate results are selected. In addition, the possibility of an entity's willingness to cede resources or it requiring additional resources is analysed period by period. The penalties for which coordination is established fall within the order of magnitude accepted by entities. This penalty limit refers to the fact that entities limit sharing their own resources in high demand situations, because they prioritise in compliance with their own requirements before allocating such resources in exchange for possible compensation. In this case, this is defined as 300 monetary units per resource unit. Finally, the assigned capacity is adjusted proportionally to the requested capacity and equals the total excess capacity.

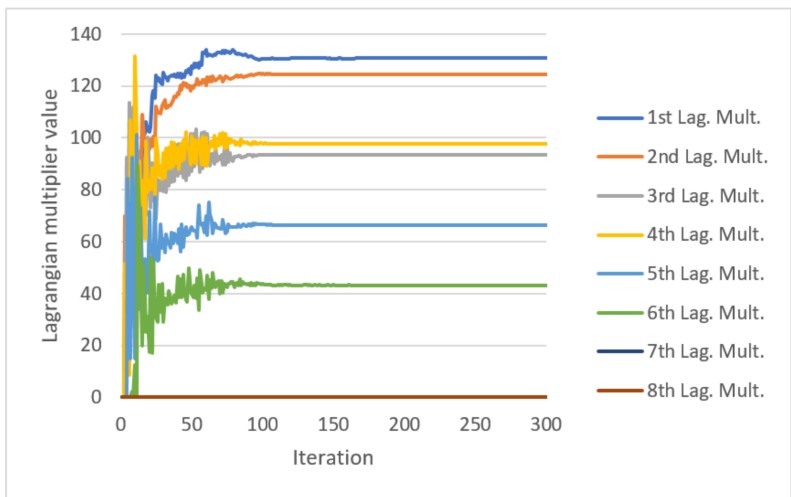

**Figure 9.** Lagrange multiplier values on the 49th planning horizon of instance STP1R30_4 coordinated with STP1R30_5 and STP1R30_6 (the STP1R30_4 Seasonal trend demand, product type P1, resource at 30%, instance 4).

It ought to be remembered that the Lagrange multiplier value increases when all the entities need the same resource, and this value is null when there is enough resource available to meet all the entities' needs. Therefore, if penalties are reasonable, entities can advance or delay their planning and compensate this with these penalty costs. Thus, entities will release the resources that another entity might more urgently need because, it is willing to assume an additional cost to avoid higher expenses. As all the entities have higher resource needs, the penalties in the Lagrange multiplier calculation iterations increase. This created situation makes entities reluctant to cede resources, because they all firstly attempt to cover their own needs. This means that no resources with high unit penalties are ceded, which is an uncoordinated situation.

Once the first proposals for the assigned resources and unit penalties are established, the entities with resource needs can recalculate their resources requirement. The established unit penalties allow additional costs to be defined for the right to use the shared resources that the entities are willing to compensate per period. Subsequently, the entities with surplus resources can recalculate their master plan by considering the resources used by other entities distributed proportionally to the capacities that they had originally reserved, and compensation for the right to use shared resources. This process is repeated on each rolling horizon.

In Figures 10 and 11, distributed coordination is observed to move closer to centralised coordination in the situations analysed with rolling horizons, and improves compared to uncoordinated entities. When analysing each combination, the median in the *TTP4R30* combination achieves even better results than the centralised coordination for the costs in Figure 10 and the service level in Figure 11.

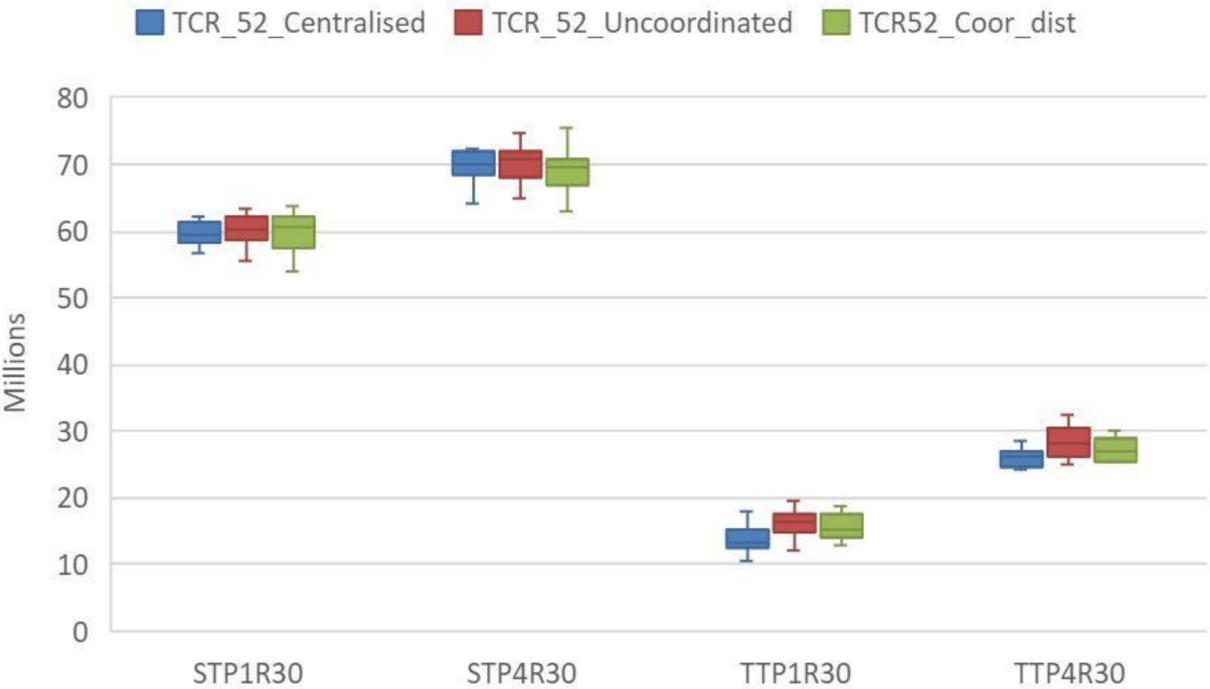

**Figure 10.** Distribution of the total costs in relation to centralised coordination, uncoordinated and decentralised coordination (the TTP1R30 trend demand, product type P1, resource at 30%; the TCR_52_Centralised total cost values for the 52 studied periods with entities sharing a resource in centralised coordination; the TCR_52_Uncoordinated total cost for the 52 studied periods with decentralised uncoordinated entities; the TCR52_Coor_dist total cost for the 52 studied periods with the implemented coordination mechanism).

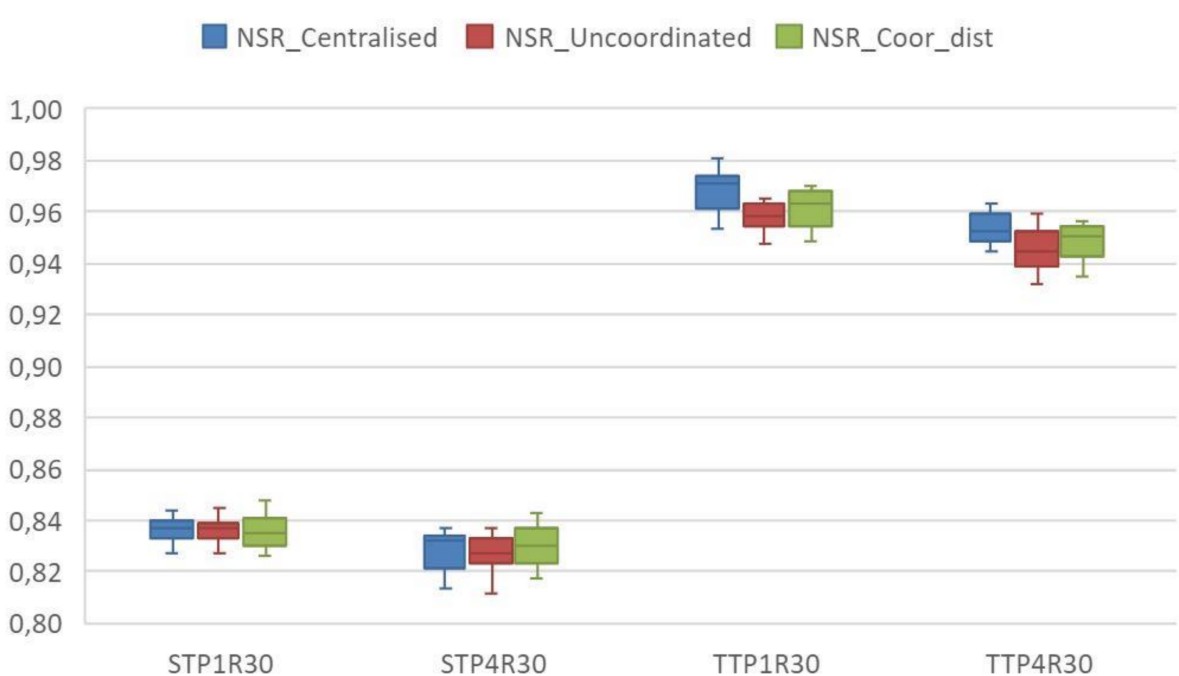

**Figure 11.** Distribution of service level in relation to centralised coordination, uncoordinated and decentralised coordination (the TTP4R30 trend demand, product type P4, resource at 30%; the NSR_Centralised service level for the 52 studied periods with the entities sharing a resource in centralised coordination; the NSR_Uncoordinated service level for the 52 studied periods with decentralised uncoordinated entities; the NSR_Coor_dist total cost for the 52 studied periods with the implemented coordination mechanism).

Figure 12 shows the distribution of the number of executed periods when capacity is shared among the entities. From this figure, we understand that, in the instances with more periods in which the entities share capacity, TTP1R30 and TTP4R30, planning with distributed coordination can achieve better results than uncoordinated planning in which no capacity is shared.

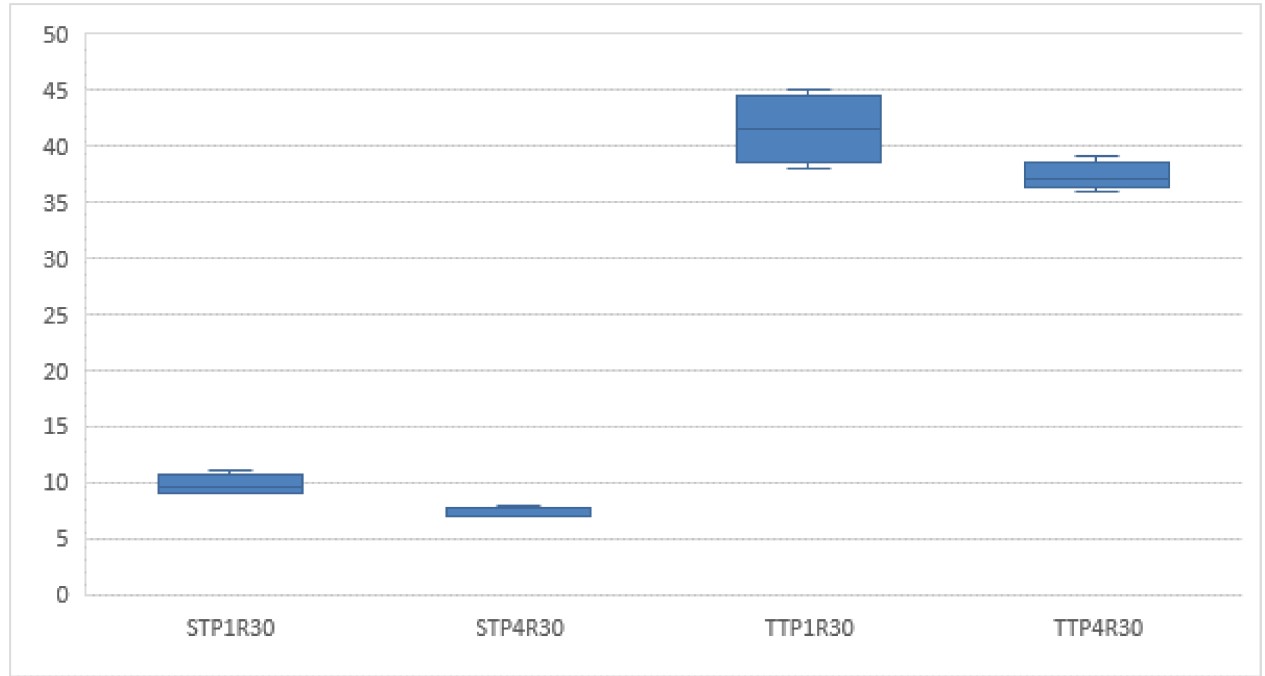

**Figure 12.** Number of executed periods where capacity is shared among the entities (source: the Authors).

An example of how capacity is shared can be seen with Figures 13 and 14. They show the behaviour of entity STP1R30_4, which shares its resource, *R1*, with STP1R30_5 and STP1R30_6. Figure 13 illustrates how entities share resources during medium demand periods. During high demand periods, entities use all their resources to meet their own demands, and do not cede any part of their available resources. During low demand periods, none of the entities needs extra resources. Entities can be observed to start from more comfortable stock situations during low demand periods, when they require fewer additional resources. It is during the high demand periods that most resources are shared. Compensations increase when the capacity requirements of the shared resources increase.

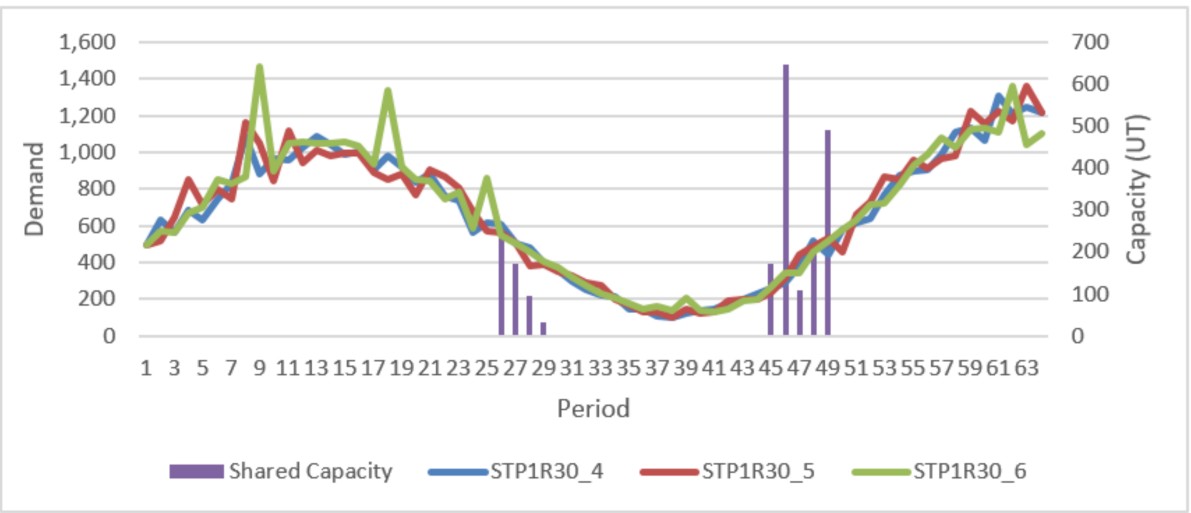

**Figure 13.** Graph of the aggregate demand for the products of entities STP1R30_4, 5 and 6 and the capacities shared among entities (STP1R30_4 seasonal trend demand, product type P1, resource at 30%, instance 4).

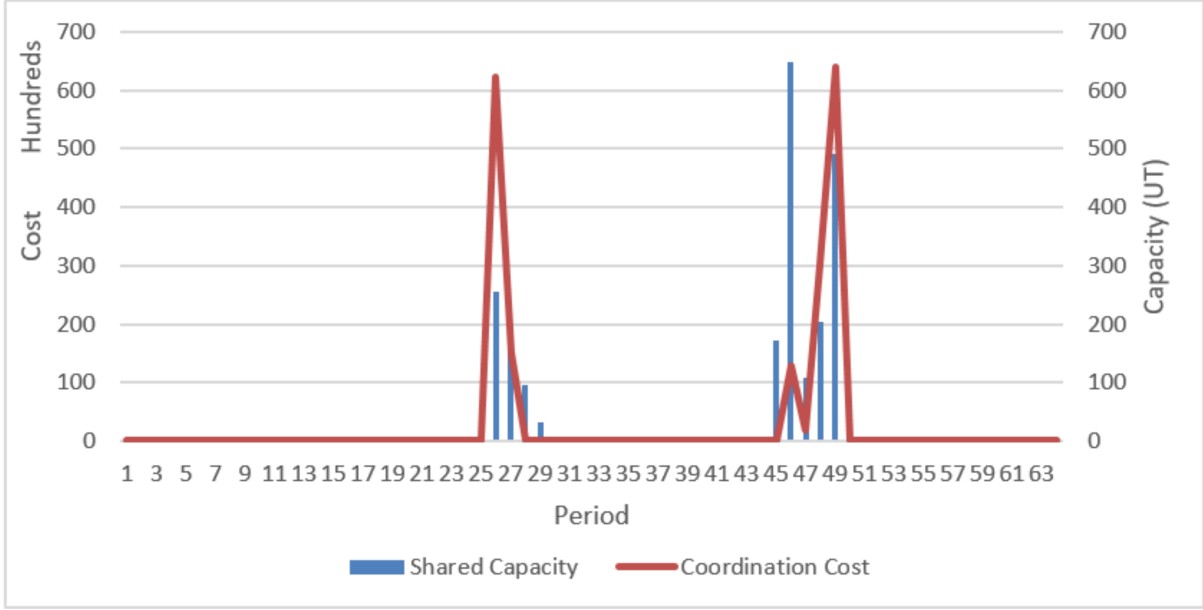

**Figure 14.** Graph of the resources shared among entities STP1R30_4, 5 and 6 during the first period and penalties (Shared Capacity, shared resources; Coordination Cost, penalties between entities).

Figures 13 and 14 show that the aggregate resource is sufficient for the specific demands during low demand periods, which means that the price for the right of use is zero. The greatest exchange of resources during the 46[th] period and penalties can be found during the 49[th] period, with increasing demand in all three entities. In the total number of

executed periods, 64 periods, the entities share capacity during 9 periods, but only during 6 of these periods are unit penalties set. During 3 of the executed periods, capacities are shared with no unit penalty for right to use.

Table 5 shows the unit penalties for the planned periods. The shaded cells denote the period during which resources are to be shared on planning horizons from 42 to 49. Entities define unit penalties for employing the shared resource for executed periods and future periods. Therefore, these planned penalties encourage entities to advance or delay their operations plans to account for increases in aggregate demand. This advance is reflected in the zero prices for the shared resources during the periods when the plan is executed. Compensations are made during the first period on each planning horizon.

**Table 5.** Penalties for the right to use shared resources among entities STP1R30_4, with 5 and 6.

|  | 42 | 43 | 44 | 45 | 46 | 47 | 48 | 49 |
|---|---|---|---|---|---|---|---|---|
| t |  |  |  |  | 20 | 16 | 147 | 131 |
| t + 1 |  |  | 11 | 10 | 44 | 49 | 129 | 124 |
| t + 2 |  |  | 2 | 5 | 53 | 25 | 103 |  |
| t + 3 |  |  |  | 10 | 11 | 32 | 87 |  |
| t + 4 |  |  |  |  |  |  | 79 | 66 |
| t + 5 |  |  |  |  |  |  | 56 | 43 |
| t + 6 |  |  |  |  |  |  |  |  |
| t + 7 |  |  |  |  |  |  |  |  |

Orange shading denotes the planned coordination periods when capacity is required from another entity. Blue shading represents the planned coordination periods when capacity is released to other entities and figures are offset.

Figure 15 shows the periods during which the entity expects to require capacity (orange shading) and the periods during which it expects to cede capacity (blue shading) for each period when it is planned for the other entity, STP130_5. Distributed planning coordinates entities in so far as they have more capacity during certain periods, and they share on different rolling horizons during others.

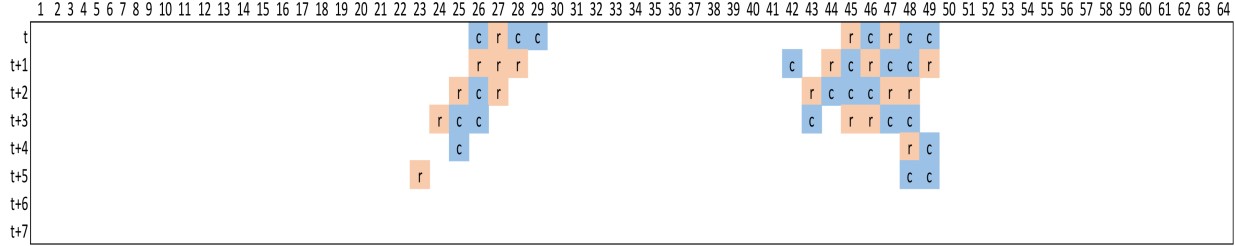

**Figure 15.** Periods with capacity shared by STP130_5 with entities STP1R30_4 and STP1R30_6 (Orange shading denotes the planned coordination periods when capacity is required from another entity. Blue shading represents planned coordination periods when capacity is released to other entities and figures are offsets).

## 6. Conclusions and Future Research

The decentralised coordination for sustainable resource sharing of independent companies by establishing an internal penalty is herein analysed by following the rolling horizons procedure on a test bed. This sustainable action reveals an average improvement in the respective entities with no need to share any internal information or an independent centralised agent. The coordination mechanism responds to SMEs' lack of resources and their information sharing distrust.

The proposed procedure allows resource sharing among independent entities with no prevailing power according to an uncoordinated situation on a rolling horizon. The operations manager seeks a plan that minimises the total costs for each planning horizon, including risk reduction and emergency preparedness for environmental and social impacts towards responsible production [15]. The method of updating Lagrange multipliers with

the subgradient method tends to stabilise in the successive iterations but does not generate a valid proposal solution, because it does not comply with the relaxed constraints. This paper presents a procedure that forces compliance with constraints by putting the stability of the internal unit penalty to the best use by ensuring resources for each entity's internal needs and by distributing the remaining resources among demanding entities.

Entities share whenever one requires resources and another has surplus. Excess resources can be generated by advancing or delaying operations to release resources. This effort made to advance operations is offset by the savings generated when this available resource is released. However, when entities' requirements are such that planning variations imply high costs, they are expected to refuse sharing their resources. Coordination occurs during periods with medium requirements and mainly during higher demand periods when some entities advance operations in exchange for compensation from others.

One of the most relevant findings appeared for those cases with greater capacity restrictions and more requirements, where the benefits of sharing resources were appreciated. For alternative processes, there are fewer resource requirements, because alternatives are available, and thus, sharing resources in these instances is not an interesting option. The proposed distributed coordination outperformed the uncoordinated situation and improved centralised coordination in some cases. Therefore, collaborative process entrepreneurs such as those herein presented can enable sustainable developments.

The presented method allows a system to share a surplus resource among entities. Moreover, entities can advance operations when the saturation of shared capacities is forecast by anticipating higher penalties during periods when the requirements of all the entities are high. This means that companies' operation planning is aligned to improve the use of shared resources with known future demand.

Therefore, the main identified impact is not having to share all information to improve a decentralised–uncoordinated situation. The coordination mechanism allows improvement in an uncoordinated situation and can even match/improve centralised operation planning, given the uncertainty and heuristics inherent to rolling horizons. Digitisation and cloud computing, which can facilitate noncritical information exchange, will enable companies to become more resilient and agile and, consequently, more sustainable in their resource management.

The proposed method presents coordination with penalties when resources are lacking for entities' aggregate requirements but has a null penalty when shared resources are sufficient. Entities may be surprised to find that they receive compensation when they share resources in certain situations but not in others. This could lead to distrust in the relationship, which should be analysed by future research. The numerical results are obtained with a combined set of entities, and future research should evaluate other combinations of entities.

**Author Contributions:** Conceptualization, G.R.-S.; methodology, G.R.-S. and J.M.; software, G.R.-S.; validation, J.P.G.-S. and J.M.; formal analysis, G.R.-S., J.P.G.-S. and J.M.; investigation, G.R.-S.; resources, G.R.-S., J.M., S.E.-M. and J.P.G.-S.; data curation, G.R.-S. and J.M.; writing—original draft preparation, G.R.-S., J.M., S.E.-M. and J.P.G.-S.; writing—review and editing, G.R.-S., J.M., S.E.-M. and J.P.G.-S.; visualization, G.R.-S., J.M., S.E.-M. and J.P.G.-S.; supervision, J.M., S.E.-M. and J.P.G.-S.; funding acquisition, S.E.-M. and J.P.G.-S. All authors have read and agreed to the published version of the manuscript.

**Funding:** This research received no external funding.

**Institutional Review Board Statement:** Not applicable.

**Informed Consent Statement:** Not applicable.

**Data Availability Statement:** The data presented in this study are available on request from the corresponding author.

**Conflicts of Interest:** The authors declare no conflict of interest.

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
