# Peer review of "Collaborative Distributed Planning with Asymmetric Information. A Technological Driver for Sustainable Development"

_sustainability, doi:10.3390/su13126628_

Round 1

Reviewer 1 Report

I was pleased to read this paper which demonstrates the authors' high degree of expertise on how to manage material flow and industrial operations in manufacturing supply chains. The study is very well done and the manuscript is written with great clarity, however before its publication there are some parts that could be improved.

  1. The literature gap stated in lines 68-69 could be turned into one or more research questions (RQs). This would help connect the theoretical framework with model validation and conclusions.
  2. In the literature review, authors should better highlight the relationships between resource use efficiency within supply chains and environmental and socioeconomic performance. As well as another issue to be explored in the literature is how an efficient system can enable sustainability, agility and organizational resilience of firms especially in times uncertainty and change. Finally, a last important topic, but not considered by the authors, is the industrial symbiosis that favors a rational use of resources.
  3. It would also be appreciated if the authors would introduce an autonomous section dedicated to the methodology, in the reading of the manuscript is felt this lack because it goes directly from the theoretical framework to the description of the mathematical model. 

Finally, as a last advice to the authors, I suggest to revise both the title of the paper and the keywords. At first glance the title does not immediately call the reader's attention to the study, which is instead relevant; while for the keywords one should pay attention to the combined use of acronyms and extended definitions.

Author Response

Firstly, the authors wish to thank the reviewer’s effort (reviewer 1) and time in reviewing the document, the quality of the comments and the support to improve the present work.

Herewith we explain our article modifications thanks to the recommendations:

  1. Comment:

The literature gap stated in lines 68-69 could be turned into one or more research questions (RQs). This would help connect the theoretical framework with model validation and conclusions.

      Action:

     Our research question has been written

Lines 107-111

This article addresses the research question of a proposed coordination mechanism for distributed collaborative operation planning between independent companies to share resources with asymmetric information and to face demand uncertainty to outperform an uncoordinated situation and a centralised situation towards sustainable development. …

  1. Comment:

In the literature review, authors should better highlight the relationships between resource use efficiency within supply chains and environmental and socioeconomic performance. As well as another issue to be explored in the literature is how an efficient system can enable sustainability, agility and organizational resilience of firms especially in times uncertainty and change. Finally, a last important topic, but not considered by the authors, is the industrial symbiosis that favors a rational use of resources.

      Action:

The introduction has been rewritten to introduce the topics

Lines 30-54

In the last decade, corporate interest in green investments has considerably increased because companies are concerned about resources efficiency and environmental issues [1], and the private sector’s commitment to collaborate [2]. This trend is a result of public policies. For example, one of the three main European Commission objectives for environmental policy is the de-coupling of resources use from economic growth through significantly improved resources efficiency, dematerialisation of the economy and waste prevention [3]. Fulfilling this goal requires synergistic changes in both policy and industry terms [4]. The sustainability concept in the supply chain management field was introduced by Carter et al. [5]. Seuring et al. [6] define sustainable supply chain management (SSCM) as the management of material, information and capital flows, and as cooperation among companies along the supply chain, while taking goals from all three sustainable development dimensions (economic, environmental, social) into account, which derive from customer and stakeholder requirements.

In order to achieve SSCM, the sustainable consumption and production topic is one of the most crucial aspects to consider. It consists in having more efficient and profitable production, using fewer raw materials and adding value to a product, while creating less pollution and waste during this process [7]. Tseng et al. [8] explain that SSCM reduces resources, material and waste by enabling better resource utilisation, which plays a significant role in achieving social, environmental and economic performance.

Industrial symbiosis is another strategy to achieve SSCM [9], which is the collective resource optimisation concept based on sharing services, utility and by-product resources among diverse industrial processes or actors to add value, reduce costs and improve the environment. The keys to industrial symbiosis are the collaboration and synergistic possibilities offered by geographic proximity, which generally focuses on the physical exchange of materials, energy, water and by-products. Industrial symbiosis could be a considerable financial benefit in raw material substitution and transportation cost savings [10].

Lines -68-77

In logistics, the potential of logistic-sharing solutions and respective transport capabilities to reduce emissions and mitigate the transport sector's impacts on climate change also implies benefits for companies by reducing overall operating expenses and transport costs per kilogram, and by cutting maintenance and personnel costs, because fewer assets are needed [17]. As Shuai et al. [18] point out, online retailers usually adopt capacity sharing to cope with the demand surge because of unmanned distribution’s low cost, especially because demand tends to be uncertain.

Sharing resources is increasingly easier thanks to digitisation [19] regardless of the cooperation level, while organisations’ increased resilience helps to deal with the complexity of change, while preserving the capacity for development [20]..

  1. Comment:

It would also be appreciated if the authors would introduce an autonomous section dedicated to the methodology, in the reading of the manuscript is felt this lack because it goes directly from the theoretical framework to the description of the mathematical model. 

      Action:

An autonomous section has been added for the Methodology

Lines 210-257

  1. Methodology

A multistep approach was followed at different demand uncertainty levels to help to improve operations planning for sustainable development.

Firstly, the GMOP formulation is presented. It helps to establish mathematical programming that contemplates the possibility of alternative operations [29,65,66]. Therefore, the adopted formulation moves closer to companies’ reality.

This section presents the modelling of the centralised coordination model through the compact model and how the model formulation also allows uncoordinated models to be considered. Afterwards, the proposed coordination mechanism that uses Lagrange relaxation is introduced. The Lagrangian multipliers calculation, which allows prices to be obtained for shared resources and the distributed coordination mechanism, is described. A flow chart (Figure 3) is presented to view the coordination mechanism proposal steps, while Figure 1 presents this first step approach.

Figure 1: Modelling approaches.

Secondly, the design of the numerical experiment is presented based on an available test bed [67]. The flow chart in Figure 2 illustrates the experiment design. This test bed allows several operations planning variables to be analysed, such as different demand types including expected variations (constant, trend, seasonal, combinations), suspected variations (noise in uniform distribution), unknown variations (uncertainty with random variation), and the irregular distribution of demand among final products. This test bed provides different product complexities and the possibility of alternative operations. These instances include variations across periods and between several planning horizons, which takes us closer to the reality of companies’ uncertainties. The test bed takes into account the need to break symmetries to facilitate resolution and to avoid instabilities in its calculation [68]. The cost structure varies according to the proximity to the decision making in each planning horizon by adjusting to the industrial reality of considering resource variations in mid- and long-term forecasts. Resources capacities have different availability levels that facilitate the analysis of instability situations because the companies' available capacity is saturated. Instances are designed to use the rolling horizon heuristic present in industry [69] with previous periods to avoid initial stocks and final periods influencing the simulating continuity objective.

For this computer simulation, 216 instances were selected to analyse the influence of various saturation levels of available resources and the impact of three product structures. These instances are considered the most susceptible because they involve using more of their resources capacities and, consequently, have more probabilities of requiring shared resources. The variables of unknown demand variation and irregular demand distribution have been left for future analysis work.

In order to evaluate different solutions, both total costs and service levels were taken as indicators of economic, environmental and social sustainability. It was assumed that employing the available resources between companies would lead to better environmental performance, cost reduction towards economic sustainability, and both would create jobs.

 Thirdly, the simulation results of mathematical programming in an uncoordinated situation versus the operation planning of centralised coordination are presented. This comparison allows those cluster families for which available capacity is lacking to be identified, and it is possible to share their resource capacity, which could be a competitive advantage. Subsequently in those families of instances for which improvements were observed, distributed operations planning was  simulated with the proposed coordination mechanism. This allowed the results of the total costs, service levels and number of periods sharing resources for each instance family to be analysed.

Figure 2: Numerical experiment design.

  1. Comment:

Finally, as a last advice to the authors, I suggest to revise both the title of the paper and the keywords. At first glance the title does not immediately call the reader's attention to the study, which is instead relevant; while for the keywords one should pay attention to the combined use of acronyms and extended definitions.

      Action:

Title and keyboards has been updated

Lines 2-4

Collaborative distributed planning with asymmetric information. A technological driver for sustainable development

Lines 25-26

Keywords: Supply chain planning; Sustainability; Lagrangian relaxation; Resources sharing; Collaborative planning; Mathematical Programming.

  1. General Comments

English language and style are fine/minor spell check required

      Action:

          The text has been rewritten and proofread by a native English translator. (Letter at the end of rebuttal)

Reviewer 2 Report

The purpose of the article is not explained in the abstract.

The methodology of empirical research should be described in more detail.

Especially the methodology of selecting the research sample should be specified.

Author Response

Firstly, the authors wish to thank the reviewer’s effort (reviewer 2) and time in reviewing the document, the quality of the comments and the support to improve the present work.

Herewith we explain our article modifications thanks to the recommendations:

  1. Comment:

The purpose of the article is not explained in the abstract.

      Action:

The abstract has been rewritten in order to clarify the purpose.

Lines 10-24

Abstract: The growing interest in sustainable development is reflected in both the market's sensitivity to environmental and social issues and companies' interest in the opportunities that sustainable development objectives provide. SMEs, which account for most of the world's pollution, have significant resource constraints for a sustainable development. Sharing their scarce resources can help them to overcome these constraints, and to gain agility and organisational resilience against uncertainties, but the distrust inherent in belonging to different companies prevents them from sharing the necessary information for coordination purposes. This paper presents a coordination mechanism proposal with information asymmetry to allow independent companies’ resources to be sustainably shared as a technological driver. The proposed distributed coordination mechanism is compared to both a decentralised-uncoordinated and a centralised situation. The interest of the proposal is evaluated by a computer simulation experiment employing mathematical programming models with independent objectives in the Generic Materials and Operations Planning formulation with a rolling horizon procedure in different demand, uncertainty and product scenarios. Competitive improvement is identified for all members for their excess capacity use and their operations planning.

  1. Comment:

The methodology of empirical research should be described in more detail.

Especially the methodology of selecting the research sample should be specified.

               Action:

An autonomous section has been added for the Methodology and the methodology of selecting the research sample has been specified.

Lines 210-257

  1. Methodology

A multistep approach was followed at different demand uncertainty levels to help to improve operations planning for sustainable development.

Firstly, the GMOP formulation is presented. It helps to establish mathematical programming that contemplates the possibility of alternative operations [29,65,66]. Therefore, the adopted formulation moves closer to companies’ reality.

This section presents the modelling of the centralised coordination model through the compact model and how the model formulation also allows uncoordinated models to be considered. Afterwards, the proposed coordination mechanism that uses Lagrange relaxation is introduced. The Lagrangian multipliers calculation, which allows prices to be obtained for shared resources and the distributed coordination mechanism, is described. A flow chart (Figure 3) is presented to view the coordination mechanism proposal steps, while Figure 1 presents this first step approach.

Figure 1: Modelling approaches.

Secondly, the design of the numerical experiment is presented based on an available test bed [67]. The flow chart in Figure 2 illustrates the experiment design. This test bed allows several operations planning variables to be analysed, such as different demand types including expected variations (constant, trend, seasonal, combinations), suspected variations (noise in uniform distribution), unknown variations (uncertainty with random variation), and the irregular distribution of demand among final products. This test bed provides different product complexities and the possibility of alternative operations. These instances include variations across periods and between several planning horizons, which takes us closer to the reality of companies’ uncertainties. The test bed takes into account the need to break symmetries to facilitate resolution and to avoid instabilities in its calculation [68]. The cost structure varies according to the proximity to the decision making in each planning horizon by adjusting to the industrial reality of considering resource variations in mid- and long-term forecasts. Resources capacities have different availability levels that facilitate the analysis of instability situations because the companies' available capacity is saturated. Instances are designed to use the rolling horizon heuristic present in industry [69] with previous periods to avoid initial stocks and final periods influencing the simulating continuity objective.

For this computer simulation, 216 instances were selected to analyse the influence of various saturation levels of available resources and the impact of three product structures. These instances are considered the most susceptible because they involve using more of their resources capacities and, consequently, have more probabilities of requiring shared resources. The variables of unknown demand variation and irregular demand distribution have been left for future analysis work.

In order to evaluate different solutions, both total costs and service levels were taken as indicators of economic, environmental and social sustainability. It was assumed that employing the available resources between companies would lead to better environmental performance, cost reduction towards economic sustainability, and both would create jobs.

 Thirdly, the simulation results of mathematical programming in an uncoordinated situation versus the operation planning of centralised coordination are presented. This comparison allows those cluster families for which available capacity is lacking to be identified, and it is possible to share their resource capacity, which could be a competitive advantage. Subsequently in those families of instances for which improvements were observed, distributed operations planning was  simulated with the proposed coordination mechanism. This allowed the results of the total costs, service levels and number of periods sharing resources for each instance family to be analysed.

Figure 2: Numerical experiment design.

  1. General Comments

Moderate English changes required

      Action:

          The text has been rewritten and proofread by a native English translator. (Letter at the end of rebuttal)

Reviewer 3 Report

I believe that the manuscript sustainability-1227036 entitled “Coordination mechanism for planning sustainable distributed operations with shared resources. A technological driver for responsible production” is an interesting work because they provide a detailed analysis, it is well written with many experiments made by the authors who offer valuable information’s. I consider that the manuscript is sustainable to be consideration for publication after minor revision.

  1. Please revise the figures numbering!
  2. Conclusion section should be rewritten and extended by summary of the most important findings as well as the impact of the finding into current state of the art.

Author Response

Firstly, the authors wish to thank the reviewer’s effort (reviewer 3) and time in reviewing the document, the quality of the comments and the support to improve the present work.

Herewith we explain our article modifications thanks to the recommendations:

  1. Comment:

I believe that the manuscript sustainability-1227036 entitled “Coordination mechanism for planning sustainable distributed operations with shared resources. A technological driver for responsible production” is an interesting work because they provide a detailed analysis, it is well written with many experiments made by the authors who offer valuable information’s. I consider that the manuscript is sustainable to be consideration for publication after minor revision.

  1. Please revise the figures numbering!

      Action:

The Figure 1 link has been added and reviewed numbering

  1. Comment:

Conclusion section should be rewritten and extended by summary of the most important findings as well as the impact of the finding into current state of the art.

      Action:

     The conclusion has been rewritten and extended

Lines 615-632

One of the most relevant findings appeared for those cases with greater capacity restrictions and more requirements, where the benefits of sharing resources were appreciated. For alternative processes, there are fewer resource requirements because alternatives are available and, thus, sharing resources in these instances is not an interesting option. The proposed distributed coordination outperformed the uncoordinated situation, and improved centralised coordination in some cases. Therefore, collaborative process entrepreneurs like those herein presented can enable sustainable developments.

The presented method allows a system to share a surplus resource among entities. Besides, entities can advance operations when the saturation of shared capacities is forecast by anticipating higher penalties during periods when the requirements of all the entities are high. This means that companies’ operation planning is aligned to improve the use of shared resources with known future demand.

Therefore, the main identified impact is not having to share all information to improve a decentralised-uncoordinated situation. The coordination mechanism allows improvement in an uncoordinated situation, and can even match/improve centralised operation planning given the uncertainty and heuristics inherent to rolling horizons. Digitisation and cloud computing, which can facilitate non-critical information exchange, will enable companies to become more resilient, agile and, consequently, more sustainable in their resource management..

  1. General Comments

English language and style are fine/minor spell check required

      Action:

          The text has been rewritten and proofread by a native English translator. (Letter at the end of rebuttal)

Reviewer 4 Report

This paper appears to be a twenty-first century restatement of the socialist calculation problem that was addressed by Lange and Taylor in the 1930s, with a response by Hayek (1945) and others. The paper discusses reallocation of resources from firms with excess resources to those who can use more, but this is exactly what market prices do, at least according to what introductory economics courses teach their students. The market is a "resource sharing mechanism" that allocates resources to those who place the greatest value on them. The paper's numerical experiments deal with finite numbers of products, but one of the things markets to do increase sustainability is enable entrepreneurs to come up with new products and production methods that are more sustainable. The paper's framework eliminates this important source of sustainability by assumption. The paper seems to come up with a central planning solution for something that decentralized markets already do. The paper talks about sharing resources, but don't SMEs already do this? For example, manufacturing firms hire trucking firms to transport their products rather than do their own trucking. They contract out for services (advertising, accounting, etc.) that are shared by many firms. Even large firms like Boeing contract out for aircraft parts, so Boeing only assembles parts that are manufactured by other companies. The paper seems to find a solution for something that is not a problem because decentralized markets already do what the paper recommends, and again referencing Hayek (1945) in a decentralized way that does not require that firms have all the knowledge that they use. The introduction says that the paper is "applied to an academic case study" but there is no actual real-world case that is studied.

The paper would be more convincing if it used an actual real-world case study to illustrate its points, and if it would explain why decentralized markets do not already do what the paper wants to do with centralized planning.

Author Response

Firstly, the authors wish to thank the reviewer’s effort (reviewer 4) and time in reviewing the document, the quality of the comments and the support to improve the present work.

Herewith we explain our article modifications thanks to the recommendations:

  1. Comment:

This paper appears to be a twenty-first century restatement of the socialist calculation problem that was addressed by Lange and Taylor in the 1930s, with a response by Hayek (1945) and others. The paper discusses reallocation of resources from firms with excess resources to those who can use more, but this is exactly what market prices do, at least according to what introductory economics courses teach their students. The market is a "resource sharing mechanism" that allocates resources to those who place the greatest value on them. The paper's numerical experiments deal with finite numbers of products, but one of the things markets to do increase sustainability is enable entrepreneurs to come up with new products and production methods that are more sustainable. The paper's framework eliminates this important source of sustainability by assumption. The paper seems to come up with a central planning solution for something that decentralized markets already do. The paper talks about sharing resources, but don't SMEs already do this? For example, manufacturing firms hire trucking firms to transport their products rather than do their own trucking. They contract out for services (advertising, accounting, etc.) that are shared by many firms. Even large firms like Boeing contract out for aircraft parts, so Boeing only assembles parts that are manufactured by other companies. The paper seems to find a solution for something that is not a problem because decentralized markets already do what the paper recommends, and again referencing Hayek (1945) in a decentralized way that does not require that firms have all the knowledge that they use. The introduction says that the paper is "applied to an academic case study" but there is no actual real-world case that is studied.

The paper would be more convincing if it used an actual real-world case study to illustrate its points, and if it would explain why decentralized markets do not already do what the paper wants to do with centralized planning.!

      Action:

-The title has been modified to reflect the decentralized operation planning of article

Lines 2-4

Collaborative distributed planning with asymmetric information. A technological driver for sustainable development

-The abstract has been rewritten in order to explain that the evaluation has been done on a computer simulation

Lines 20-23

The interest of the proposal is evaluated by a computer simulation experiment employing mathematical programming models with independent objectives in the Generic Materials and Operations Planning formulation with a rolling horizon procedure in different demand, uncertainty and product scenarios...

-the “academic case study” has been substituted.

Lines 114-117

This approach is applied to an extensive test bed that represents a cluster of companies with no prevailing power that voluntarily decide to share the capacity of one of their resources, e.g. transport, or batch processing ovens (welding, annealing, vulcanisation, etc.), towards a more sustainable supply chain..

-the introduction has been modified in order to let our article to be in the level of the industrial symbiosis where the coordination mechanisms could be applied.

Lines 48-54

Industrial symbiosis is another strategy to achieve SSCM [9], which is the collective resource optimisation concept based on sharing services, utility and by-product resources among diverse industrial processes or actors to add value, reduce costs and improve the environment. The keys to industrial symbiosis are the collaboration and synergistic possibilities offered by geographic proximity, which generally focuses on the physical exchange of materials, energy, water and by-products. Industrial symbiosis could be a considerable financial benefit in raw material substitution and transportation cost savings [10].

-The conclusion has been rewritten in order to identify the technological requirements in order to be able to apply in the industry the proposed coordination mechanism to manage the resource.

 Lines 630-632

Digitisation and cloud computing, which can facilitate non-critical information exchange, will enable companies to become more resilient, agile and, consequently, more sustainable in their resource management..

  1. General Comments

English language and style are fine/minor spell check required

      Action:

          The text has been rewritten and proofread by a native English translator. (Letter at the end of rebuttal)

Round 2

Reviewer 1 Report

Dear Authors,

I have carefully read the new version of your manuscript and appreciate the many improvements you have introduced following the reviewers' suggestions. Therefore, I believe that now the paper is suitable for its publication in this journal.

Congratulations!